# Interpreting convolutional neural networks to study wide-field amacrine cell inhibition in the retina

**Michaela Vystrčilová,[1,*] Shashwat Sridhar,[2,3] Max F. Burg,[1] Tim Gollisch,[2-5] Alexander S. Ecker[1,6,†]**

[1] Institute of Computer Science and Campus Institute Data Science, University Göttingen, Germany
[2] University Medical Center Göttingen, Department of Ophthalmology, Göttingen, Germany
[3] Bernstein Center for Computational Neuroscience Göttingen, Göttingen, Germany
[4] Cluster of Excellence "Multiscale Bioimaging: from Molecular Machines to Networks of Excitable Cells" (MBExC), University of Göttingen, Göttingen, Germany
[5] Else Kröner Fresenius Center for Optogenetic Therapies, University Medical Center Göttingen, Göttingen, Germany
[6] Max Planck Institute for Dynamics and Self-Organization, Göttingen, Germany
[*]michaela.vystrcilova@uni-goettingen.de
[†]ecker@cs.uni-goettingen.de

## Abstract

Wide-field amacrine cells (ACs) play a unique role in retinal processing by integrating visual information across a large spatial area. Their inhibitory influence has been implicated in multiple retinal functions such as differential motion detection and the suppression of retinal activity during eye movements. However, a coherent understanding of their general function is lacking due to difficulties in directly recording from these cells and identifying effective visual stimuli to activate them. In this study, we used convolutional neural networks (CNNs) to investigate wide-field inhibition mediated by wide-field ACs in the marmoset retina. We trained CNNs to mimic the function of the retina by predicting retinal ganglion cell (RGC) population responses to naturalistic movie stimuli and optimising the most exciting inputs (MEIs) to visualise RGCs' receptive field (RF) structures. We then optimized suppressive surrounds beyond classical RGC RF boundaries, intended to capture the inhibitory effect of wide-field ACs on RGC activity. These optimized surrounds reduced MEI-elicited activity by 10% to 30%, demonstrating that CNNs not only mimic retinal responses but can also reveal hidden computational aspects of wide-field inhibition. However, suppression strength and generalization varied across architectures and datasets, indicating potential model-specific effects, highlighting the importance of cautious interpretation. Overall, our approach illustrates how interpretability methods applied to artificial neural networks can offer new hypotheses regarding biological retinal computation, paving the way for targeted experimental validation. The code is available here.

# 1   Introduction

Wide-field amacrine cells (ACs) are a diverse class of narrowly stratified inhibitory cells extending over long distances in the retinal plane, facilitating inhibition of retinal ganglion cells (RGCs) beyond the classical receptive field (RF) [1–5]. Their ability to gather information across long ranges makes them a prime suspect in differential object motion sensitivity [6, 7] and global shift suppression [8]. Subtypes such as A17, A1, and "wiry" ACs have been studied within the context of specialized circuits [9–12]. Yet, recurring connectivity motifs suggest that wide-field ACs may also play a more general computational role [13], such as shaping inhibitory surrounds in bipolar and ganglion cells or contributing to motion anticipation [14, 15].

Despite their significance, wide-field ACs are difficult to study directly. Their location deep within retinal tissue, sparse distribution, morphological diversity, and lack of targeted stimuli make recordings technically challenging. As a result, it remains unclear how wide-field ACs contribute to population-level computations and what stimuli effectively drive their activity.

Classical models have struggled to bridge this gap. Hierarchical LN-LN models, which attempt to reconstruct retinal function through combinations of linear filtering and nonlinear transformations [16–20], can include hidden elements that explicitly capture AC computations. However, due to the highly nonlinear nature of the modeled function, they are prone to overfitting and are often poorly constrained by recordings from single cells. Consequently, models that describe AC functions often rely on parameter values derived from anatomical knowledge rather than direct inference from experimental data [9, 21].

Deep learning models offer an alternative path. CNNs have been shown to approximate the nonlinear input–output functions of RGCs with high accuracy, particularly when trained on responses from populations of simultaneously recorded neurons [22–24]. Moreover, CNN models have successfully captured complex nonlinear features of retinal circuitry – such as contrast adaptation and omitted stimulus response – even without being explicitly trained to do so [25, 26]. Based on this capacity for implicit inference, we hypothesized that CNNs trained to predict RGC responses could internalize circuit-level influences from wide-field ACs and serve as a proxy for indirectly investigating them, despite having no explicit representations of ACs.

Therefore, we trained a CNN model on RGC responses to natural movies [27]. Using it as a proxy, we based our indirect wide-field AC influence investigation strategy on feature visualization methods that manipulate input using gradient descent optimization [28, 29]. We systematically optimized the input stimuli in a two-step approach, similar to that developed by Fu et al. [30]. Because inhibition can be observed only indirectly by measuring reductions in an existing baseline neural activation, we first drove the model's response to elicit activity. Then, we probed the inhibitory mechanisms to suppress the drive. We elicited the baseline activity by optimizing the most exciting input (MEI) for each RGC – maximizing its activation as predicted by the pre-trained CNN. In the second step, we held the MEI fixed and optimized the input in the region outside of the MEI. This time, we aimed to minimize the cell's predicted activation. This approach revealed spatial patterns that most effectively suppressed the MEI-driven activity, synthesizing model-inferred inhibitory surrounds. By setting the boundary between the MEI region (which is not optimized in the second step) and the surround region (which is optimized in the second step) far beyond the reach of the dendritic field of the RGC's RF, we ensured the inhibitory modulation was beyond traditional RF boundaries, affecting areas only wide-field ACs can reach.

This reverse-engineering approach consistently revealed a wide-field suppressive effect in the model-generated surrounds, typically reducing RGC responses by 10–30 %. Cells with higher firing thresholds – those harder to excite – were more effectively suppressed, suggesting a functional link between excitability and susceptibility to wide-field AC-mediated wide-field inhibition. We further tested the generalization of these surrounds across different architectures and varying training data compositions, finding that some suppressive structures generalized more broadly than others, suggesting greater biological plausibility. Collectively, these findings generate targeted hypotheses and inform experimental designs aimed at directly testing targeted hypotheses about wide-field inhibitory stimuli in biological retinas.

## 2 Results

We used a dataset of responses from marmoset RGCs, recorded with multi-electrode arrays during presentations of naturalistic movies across two retinas [27, 31], referred to throughout the paper as Retina 1 and Retina 2. From this dataset, we analyzed only reliably classified OFF midget, OFF parasol, and ON parasol cells as they were the most numerous and clearest cell clusters. The visual stimulus consisted of non-repeating "training" and repeating "test" video segments. A non-repeating segment (300 s) followed by a repeating segment (60 s) formed one trial. Each retina was presented with 20 such trials. We trained models on neuronal responses to the training video segments. The final performance was computed as the correlation between

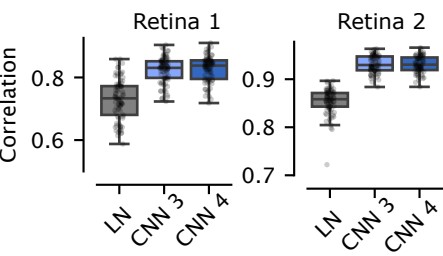

Figure 1: Performance comparison between single-cell LN models and CNNs.

model-predicted and trial-averaged spike counts from the test segments. Spike counts were binned at 85 Hz to match the video frame rate.

### 2.1 CNNs accurately capture RGC responses

To investigate the wide-field inhibition of RGCs and the structure that drives it, we used trained CNNs as proxies for the nonlinear retinal function. The CNN architectures had a *core-readout* structure, as previously used for RGC and V1 modeling [32–34]. We trained two CNN variants (CNN 3 and CNN 4, the number refers to the number of convolutional layers) per retina. The models differed in depth and convolutional parameters (Supplementary Table 1) to allow for architectural diversity, which later enabled us to test generalization. The CNNs' RFs covered a broad spatial area (900–1200 µm) for each cell, ensuring that they could capture both classical RGC RFs and extended surround influences from wide-field ACs. The classical RGC RF size is below 400 µm [14, 35].

To confirm the CNNs as appropriate models for our analyses, we compared them to a baseline: single-cell linear-nonlinear (LN) models trained on the same data [36]. The CNNs outperformed the baseline models significantly. On Retina 1, the LN model's average correlation was 0.7 ($\pm$ 0.1, standard deviation) across the classified cells (Figure 1, left). For the CNNs, the mean correlation was 0.81 ($\pm$ 0.5) for CNN 3 and 0.82 ($\pm$ 0.5) for CNN 4, averaged across cells and four models with different initializations per architecture. On Retina 2, the averages for all models were higher. The LN model averaged 0.85 ($\pm$ 0.03) correlation (Figure 1, right), and both CNNs 3 and 4 had an average correlation of 0.93 ($\pm$ 0.02).

### 2.2 MEIs generalize across architectures and elicit stronger responses than linear filters

We used feature visualization [28, 29] on our frozen pre-trained CNNs to study the wide-field AC inhibition within the retinal circuit (Figure 2A). We first needed to elicit activity in the center of the cell's RF, so that the surround could inhibit this activity. Therefore, we first optimized an MEI for each RGC by maximizing the CNN-predicted value for this cell (Figure 2B, left). Subsequently, we fixed an area around the MEI, re-initialized the surround around this fixed area, and then optimized only the surround using gradient descent, this time to minimize the cell's predicted activity (Figure 2B, right). This allowed us to isolate the contribution of the wide-field inhibitory mechanisms in the model. The center of the RF contained a fixed, strongly excitatory stimulus – the MEI. By minimizing the cell's output through optimization of only the surrounding area, we identified spatial patterns that most effectively suppressed this central excitation – revealing the structure driving wide-field inhibition as learned by the model.

We first verified that the MEIs strongly drive the model. We compared their predicted activations to those of LN filters estimated for the same cells with the same contrast. Although both structures looked similar, MEIs consistently drove higher predicted activations (Figure 2C, left). To confirm that MEIs were not overfitted to specific architectures, we tested whether MEIs generated by one CNN also activated the other: an MEI from CNN 3 was fed to CNN 4 and vice versa. We observed consistent activation across architectures (Figure 2C, right), showing that the models converge on similar excitatory features.

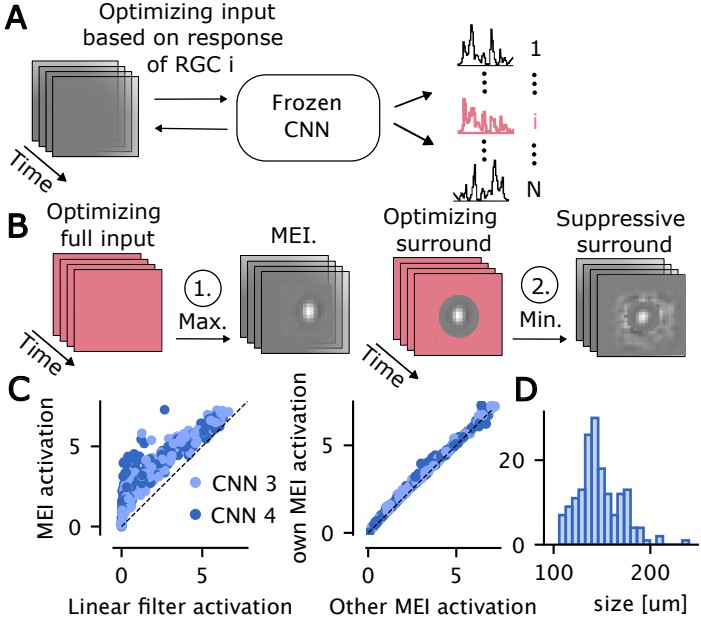

Figure 2: **A.** Schematic of the input optimization procedure: given a frozen CNN ensemble trained to predict RGC responses, spatiotemporal inputs are optimized to either maximize or minimize the predicted activation of a target RGC $i$. **B.** Two-step optimization procedure: In step 1 (left), the most exciting input (MEI) is obtained by optimizing the full input to maximize the model's prediction for the selected cell. In step 2 (right), the surround region is optimized to minimize the cell's response, while the MEI center is fixed. **C.** Left: Comparison of MEI and linear filter activations across cells, showing consistently stronger activation from MEIs across two architectures (CNN 3 and CNN 4). Right: Comparison of the activation predicted by a CNN for an MEI generated on the CNN architecture itself (y-axis) and the other CNN architecture (x-axis). **D.** Distribution of RGC RF sizes estimated from MEIs.

## 2.3 Surround optimization reveals inhibitory structures outside the classical RF

To investigate the wide-field inhibitory effects in our models, we optimized the surround regions of the input to minimize predicted activity while keeping the excitatory MEI center fixed. The goal was to identify spatial patterns in the periphery that suppress the RGC activity – mimicking the influence of wide-field ACs. To ensure that the suppressive surround optimization does not intrude on the classical RGC RF, we defined a conservative boundary. Based on prior estimates of marmoset RGC dendritic fields [35], excitatory conductance centers [14], and our own MEI-based RF size estimates (100–250 μm, Figure 2D), we used a 360-μm circular diameter protective zone for the MEI and optimized only the area outside this protective zone in this second step.

To quantify the effect of the surrounds, we used two metrics: **self-suppression** and **cross-suppression**. Self-suppression was the portion of the MEI-elicited CNN activation suppressed by a surround optimized using the same CNN architecture. Cross-suppression was the portion of the MEI-elicited CNN activation suppressed when using an MEI and a surround that were optimized using a different CNN architecture. We formally define suppression $s$ for cell $c$ on architecture $a$ given architecture $b$ as:

$$s_{c,a,b} = 1 - f_{c,a}(M_{c,b} + S_{c,b})/f_{c,a}(M_{c,b}); \qquad (1)$$

where $f_{a,c}$ is the function of CNN architecture $a$ for cell $c$, $M_{c,b}$ is the MEI estimated by architecture $b$ for cell $c$ and $S_{c,b}$ is the suppressive surround estimated by architecture $b$ for cell $c$. Self-suppression is suppression when $a = b$, cross-suppression when $a \neq b$. Cross-suppression indicates architecture-independent surround effects; large gaps between self-suppression and cross-suppression suggest overfitting.

We found significant suppressive effects of the optimized wide surrounds (Figure 3A). On Retina 1, the average self-suppression was 60 % (± 12 %, standard deviation across neurons) and 47 % (±

15 %) for CNN 3 and CNN 4, respectively. On Retina 2, the average self-suppression was 22 % ($\pm$ 10 %) and 28 % ($\pm$ 12 %) for CNN 3 and CNN 4, respectively. The cross-suppression was substantially lower for both retinas. On Retina 1 the cross-suppression of a surround generated by CNN 4 applied to CNN 3 averaged over cells, i.e., $s_{3,4}$ was 29 % ($\pm$ 11 %) and $s_{4,3}$ was 24 % ($\pm$ 10 %). On Retina 2, the averages were closer to the average self-suppression. $s_{3,4}$ was 14 % ($\pm$ 5 %) and $s_{4,3}$ 17 % ($\pm$ 5 %).

Self-suppression was consistently stronger than cross-suppression, implying a certain level of overfitting to the specific architecture, especially on Retina 1. Nevertheless, the cross-suppression effect was consistently between 10 % and 30 %, indicating learned wide-field inhibition that generalized across architectures. For further analyses, we used cross-suppression as the measure of suppression.

We examined the relationship between suppression and other cell properties. The strongest correlation we found was with a learnable parameter $\delta$ of a learned LN model's[1] softplus nonlinearity, which had the form $f_{\text{LN}}(x) = \alpha \log(1 + e^{\gamma x - \delta})$ where $\alpha, \gamma$, and $\delta$ were learnable parameters and $x$ was the input stimulus convolved with the LN model filter. The $\delta$ parameter can be interpreted as the firing threshold of a cell. The higher $\delta$, the higher the threshold for firing and thus, the harder it is to excite the cell (Figure 5B inset). Higher $\delta$ values correlated with higher suppression (Pearson's correlation coefficient of $0.55$), i.e., cells that were harder to excite were also more successfully suppressed (Figure 3B). The cross-suppression also correlated negatively with the mean firing rate (Pearson's correlation coefficient of $-0.64$), further supporting that cells that are harder to excite are also more substantially suppressed (Supplementary Figure 2B).

## 2.4 Suppression differs by cell type and retina

To better understand the effects of surround suppression, we examined whether the suppressive effects differ by cell type and across retinas. The two retinas in our dataset contained different sets of RGC types: Retina 1 included reliably identified OFF midget and OFF parasol cells but lacked ON cells, whereas Retina 2 contained OFF and ON parasol cells but no reliable midgets.

Qualitative inspection of the suppressive surround patterns revealed systematic differences. In Retina 1, the surrounds of OFF parasols and OFF midgets, as well as ON parasols in Retina 2, often showed a loosely regular tiling of light and dark patches, aligned with the polarity of the cell's center (Figure 3D). In contrast, OFF parasols in Retina 2 exhibited smoother surrounds with a distinct OFF region at the boundary of the masked MEI area. The highest pixel intensity of the suppressive surround was temporally slightly ahead of the MEI's spatial profile for all cell types. The high-saturation patches in the surround displayed a temporal ON-OFF or OFF-ON polarity progression for OFF and ON cells, respectively.

On Retina 1, quantitative analysis by cell type revealed that the cross-suppression for both OFF midget and OFF parasol cells was almost identical, averaging around 26% (Figure 3C, light green, brown). In contrast, on Retina 2, we observed significant differences between the suppression of OFF and ON parasol cells (Figure 3C, dark green, yellow). The MEI-elicited activation of ON parasols was, on average, suppressed by 20% across the two architecture types, whereas for OFF parasols, it was only 11%. Notably, OFF parasols, present in both retinas, differed markedly in suppression: 26% in Retina 1 versus 11% in Retina 2. This divergence pointed to differences not explained by cell type alone.

## 2.5 Suppression is partly architecture specific

Given the inconsistent suppressive effects between retinas and cell types, we sought to probe whether the suppressive effects were robust or model-dependent. To test the extent to which the generality of the surrounds depends on architectural choices of the CNN, we trained two additional (group B) CNN architectures (CNN 3B and 4B) with reduced capacity, using grouped convolutions and no batch normalization (Supplementary Table 1). These models performed only slightly worse than the original CNNs and still significantly outperformed the LN baseline (Supplementary Figure 1A).

We repeated the MEI and surround optimization for group B models using the same procedure as for the initial group. First, we generated MEIs and checked whether they generalized both within

---

[1]The LN models for this analysis were taken from [27] and not from [36] because [27] normalized the LN models' filter values and thus the non-linearity parameters were more informative.

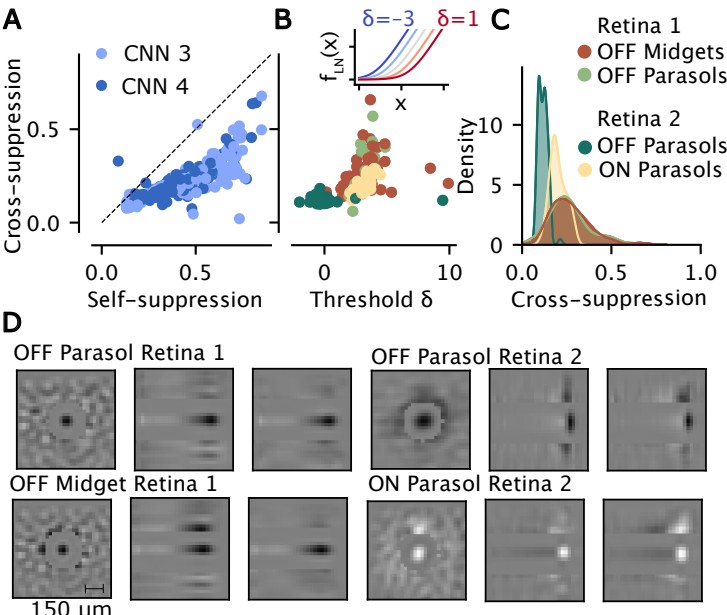

Figure 3: **A.** Cross-suppression versus self-suppression for individual cells. Each dot represents a cell; colors indicate CNN ensemble type (3 or 4). Self-suppression is computed within each architecture; cross-suppression reflects suppression when applying surrounds from one architecture to another. Color reflects the architecture to which the surround was applied. **B.** The relationship between cross-suppression (y-axis) and the firing rate threshold $\delta$ (x-axis). Colored by cell type (as in C). The effect of the threshold $\delta$ (when $\alpha = 1$ and $\gamma = 1$) on the shape of the softplus nonlinearity is included as an inset. **C.** Cell-type specific cross-suppression distributions. **D.** Visualization of MEIs and their corresponding suppressive surrounds for selected example cells from each cell type. Within each cell type, the first panel shows the MEI and suppressive surround six frames before the spike, the second panel shows a vertical spatial cut through the center of the MEI and suppressive surround over time, and the third shows a horizontal cut through the center of the MEI + suppressive surround over time (time on x-axis in both). Scale bar: 150 μm.

and across the groups. This means, for example, whether MEIs generated on CNN 3 drive CNNs 4 (within group) and drive CNNs 3B and 4B (across groups) as much as their own MEIs. We found that even though the absolute activations predicted by group B CNNs were lower than the activations predicted by CNNs 3 and 4, MEIs generalized across all architectures (Supplementary Figure 1B).

Secondly, we generated suppressive surrounds and evaluated the consistency of the suppressive effects by measuring cross-suppression within and across architecture groups. For example, for CNN 3, the cross-suppression within group was the suppressive effect that surrounds generated by CNN 4 had on the activation of CNN 3. The cross-suppression across groups was the average suppressive effect that the surrounds generated on CNNs 3B and 4B had on the activation of CNN 3.

On Retina 2, cross-suppression values were consistent both within and across architecture groups (Figure 4, right, purple, and blue), indicating robust surround patterns. On Retina 1, in contrast, cross-suppression dropped substantially when comparing across

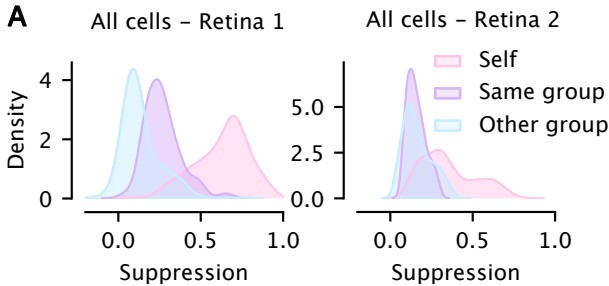

Figure 4: Each panel shows distributions of self-suppression (pink) and cross-suppression when using surrounds generated from an architecture within-group (purple) and across-group (blue). The displayed distributions are averages of the corresponding distributions across all four architectures – CNN 3, 4, 3B, and 4B.

groups (to 4–10 % on average), while remaining stronger within groups (Figure 4, left, purple and blue). This pattern suggests that suppressive surrounds in Retina 1 are more sensitive to the specifics of model architecture.

## 2.6 Cell type differences

To investigate why surround generalization was successful on Retina 2 and less so on Retina 1, we examined how these differences are reflected across cell types. Specifically, we asked whether generalization holds for individual cell types across architecture groups. We found that in Retina 1, both OFF midgets and OFF parasols were more suppressed by surrounds generated within the same model group than by those from a different group (Figure 5A, panels one and two). In contrast, in Retina 2, both ON and OFF parasols were equally suppressed by within- and across-group surrounds (Figure 5A, panels three and four). The stronger suppression of ON parasols compared to OFF parasols in Retina 2 was also preserved across architecture groups. Thus, the difference did not seem to stem from cell types but more likely from the differences between the two retinas.

Next, we tested whether the reliability of neural responses could explain the generalization differences. We ranked cell types by their reliability – measured as their explainable variance [37]. ON parasols in Retina 2 were most reliable, followed by Retina 1 OFF parasols, Retina 2 OFF parasols, and finally Retina 1 OFF midgets (Supplementary Figure 2A). However, reliability did not directly predict generalization. While the surrounds of ON parasols generalized well and ON parasols had high reliability, for Retina 1 OFF parasols – despite being more reliable than Retina 2 OFF parasols – the surrounds did not generalize. This suggests that cell-type-specific reliability does not account for the observed patterns.

Despite cell-type-specific reliability not explaining the observed patterns, we hypothesized that the overall reliability of a retina may be a factor in generalization. The lower reliability of OFF midgets in Retina 1 could influence the CNN's training on this retina, such that Retina 1 OFF parasols also overfit. In contrast, the high-reliability ON parasols in the training data of Retina 2 could improve the generalization abilities of the Retina 2 OFF parasol surrounds.

We also considered an alternative hypothesis: that the presence of both ON and OFF cells in the training set is necessary for learning inhibitory surrounds that generalize. Since Retina 2 included both polarities and Retina 1 only included OFF cells, we hypothesized that polarity diversity might be essential.

To test both hypotheses, we trained all CNN architectures (CNN 3, 4, 3B and 4B) for each cell type, using only cells of that specific cell type. We treat OFF parasols from Retina 1 and OFF parasols from Retina 2 as different cell types. We then assessed whether suppressive surrounds from these cell-type-specific models generalized. If reliability was the key factor, surrounds of Retina 1 OFF parasols (high reliability, no influence from low-reliability OFF midgets) should generalize, and Retina 2 OFF parasols (lower reliability, no influence from high-reliability ON parasols) should not. If polarity diversity were key, neither Retina 1 nor Retina 2 cell-type-specific models should generalize.

We first checked the performance of the cell-type-specific CNNs. Predictive performance remained comparable to the full models (Supplementary Figure 3), with no substantial loss in accuracy. We then examined the suppressive effects of the surrounds and generalization across architectures. In Retina 2, both ON and OFF parasols retained stable suppression and generalized across architecture groups, even in the single-cell-type models (Figure 5B, panels three and four). In Retina 1, OFF parasols showed improved generalization compared to the full model (Figure 5B, panel two), while OFF midgets showed stronger suppression but no improvement in generalization (Figure 5B, panel one).

These findings allowed us to rule out the polarity-diversity hypothesis: generalization did not depend on having both ON and OFF cells in the training data. The reliability hypothesis was only partially supported. While OFF parasols from Retina 1 improved in generalization when isolated from OFF midgets, generalization in Retina 2 persisted despite lower OFF parasol reliability. This suggests that a minimal level of reliability is likely necessary, but not sufficient, for generalization.

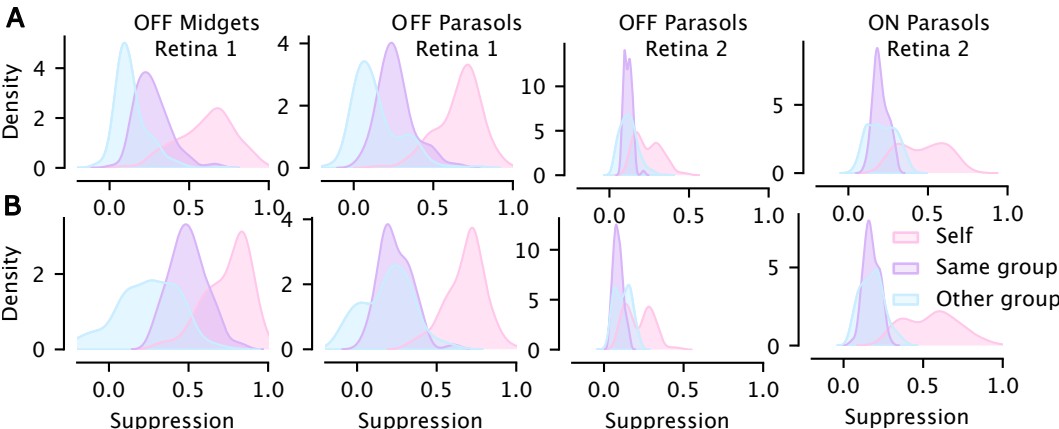

Figure 5: **A.** Each panel shows distributions of the self-suppression (pink) and cross-suppressions when using surrounds generated from an architecture within-group (purple) and architectures across-groups (blue). The displayed distributions are averages of the corresponding distributions across all four architectures – CNN 3, 4, 3B, and 4B. The architectures for each panel were trained on all reliable cells – same as those in Figure 4C. **B.** Same as **A.**, but the CNN architectures generating the surrounds were trained on cells of only the displayed cell type.

Taken together, these experiments indicate that neither polarity balance nor cell-type reliability alone can fully explain the variation in surround generalization. Further analyses and experiments would be necessary to establish generalization criteria for suppressive surrounds.

## 3 Discussion

Deep learning models have become a central tool in systems neuroscience for studying neural computation. When such models are sufficiently accurate, they are often used as functional proxies for neural circuits, enabling the generation of hypotheses that can guide experimental work [28–30, 33, 38]. Following this approach, we trained CNNs to predict RGC responses to naturalistic movies, and treated these models as proxies for retinal function in order to study inhibitory mechanisms, in particular, wide-field inhibition mediated by wide-field ACs.

To investigate this form of inhibition, we applied feature visualization to the trained CNNs. Using a two-step procedure, we first optimized MEIs that maximally activated each RGC model, and then optimized the surrounding stimulus region (with the MEI held fixed) to minimize model-predicted activity. The optimized surrounds reflected spatial patterns that the model associated with inhibition beyond the classical receptive field – potentially capturing the effects of wide-field ACs.

This approach revealed suppressive surrounds that reduced MEI-elicited activation. In the absence of direct experimental validation, we used cross-architecture generalization as an in silico sanity check for model-derived surrounds. If the suppressive patterns reflected meaningful aspects of wide-field inhibition, one would expect them to be robust across independently trained models. However, we observed that the reliability and generality of inhibition varied by dataset. In Retina 1, suppressive surrounds were strong within the trained architecture itself (up to 60 % self-suppression), but generalized poorly across CNN variants, with cross-suppression between architectures dropping to 20 %, and across architectural groups to 4–10 % (Figure 4). This raises concerns about architecture-specific overfitting. In contrast, Retina 2 showed weaker suppression (10–30 %), but much stronger cross-architecture generalization, including across architectural groups. Additionally, in Retina 2, ON parasol cells exhibited stronger suppression than OFF parasols, suggesting a degree of biological specificity and generally higher biological plausibility of the model's inferred representations (Figure 3C,D).

**Contextualizing suppressive surrounds with retinal circuitry** The suppressive surround features we identified can be related to several biological aspects of retinal circuitry. First, we observed a correlation between stronger surround suppression and higher firing thresholds across both retinas (Figure 3B), suggesting that wide-field AC-mediated inhibition may contribute to elevating thresholds

in certain RGCs. Such threshold-raising effects of ACs have been documented in the salamander retina [39], though distinctions between wide- and narrow-field ACs were not made, and their relevance to primate retinas remains unresolved.

Second, the observation that OFF midgets are suppressed similarly to OFF parasols in Retina 1 and more compared to OFF parasols in Retina 2, is surprising in light of anatomical and physiological studies indicating that parasols should be more strongly influenced by wide-field inhibition than midgets [40–43]. To disentangle whether these effects arose from biology or model training, we trained CNNs on individual cell types. This analysis revealed that Retina 1 OFF parasols generalized better across architectures when isolated from OFF midgets, whereas OFF midgets themselves showed stronger suppression but no improvement in generalization. These results might suggest that the midget surrounds are the least trustworthy: they do not fully generalize under any tested conditions, thus probably reflect model-driven artifacts rather than reliable wide-field inhibitory mechanisms.

Finally, when comparing OFF parasols across the two retinas, we observed differences that extend beyond suppression strength. In Retina 1, OFF parasols exhibited stronger suppression than in Retina 2 (Figure 3C) and showed surrounds more similar to those of OFF midgets and ON parasols (with reversed polarity) than to OFF parasols in Retina 2 (Figure 3D). Shifts in nonlinearity were also reported by [27], who analyzed the same dataset, paralleling these differences: Retina 1 OFF parasols displayed more nonlinear behavior and a higher firing threshold, while Retina 2 OFF parasols had lower firing thresholds and were more linear in their response (Figure 3B). Retina 1 OFF parasols were also significantly slower compared to Retina 2 parasols (Supplementary Figure 2C). These findings indicate that the divergence between OFF parasols across datasets is not limited to suppression magnitude but also includes other biological properties. Experimental factors such as temperature [44] or eccentricity [7] may play a role, although it is unclear how they should affect wide-field inhibition [45].

**The biological plausibility of the surrounds** The key question here is whether the suppressive surrounds identified through our feature-visualization-based approach reflect genuine mechanisms of wide-field inhibition in the biological retina. Several findings from our analyses provide compelling support for biological plausibility. Notably, the robust generalization of suppression effects across different CNN architectures in Retina 2 suggests the identification of genuine inhibitory motifs rather than artifacts of model-specific training. Moreover, the general correlation between RGC firing thresholds and the strength of suppression shows alignment with biological principles, reinforcing the potential relevance of these model-inferred surrounds to real retinal circuitry.

However, the weaker generalization observed in Retina 1 indicates a degree of architecture-specific overfitting, highlighting limitations of directly translating these findings into biological interpretations without experimental validation.

That said, the lack of generalization does not rule out biological validity. It is still possible that all model-generated surrounds would suppress RGC activity in the retina, and that the limitations lie with the models. Prior studies have successfully used CNNs to generate and experimentally confirm hypotheses about neural circuits without studying cross-architectural consistency [28, 30, 33, 38]. The generalization tests performed in this study offer a potentially useful heuristic in the absence of direct experimental validation. However, even the interpretive value of our approach ultimately depends on how well it corresponds to outcomes from biological experiments.

# 4 Methods

**Marmoset data** We used data from two adult common marmoset retinas [27], which contained RGC responses to naturalistic movie stimulation recorded using multi-electrode arrays (MEAs). The naturalistic movie stimulus was an adapted version of a short, publicly available science-fiction film by the Blender Foundation, Tears of Steel[2]. It was jittered to mimic natural eye saccades and fixation statistics. The stimulus was shown at a frame rate of 85 Hz. The experiment consisted of 20 trials, each trial containing a 300-second segment of repeating frames used for training and validation, and a 30-second segment used for final evaluation.

---

[2]https://mango.blender.org/

**CNN Architecture** We trained two groups of CNN models with distinct architectural features – CNN 3 and 4, and CNN 3B and 4B (Supplementary Table 1 for detailed differences). All models followed a similar core-readout structure as in [34]. The core consisted of three or four layers of spatiotemporally separable convolutions, each followed by an Exponential Linear Unit (ELU) nonlinearity [46]. The 3 and 4 architectures additionally included batch normalization [47] between layers, while group B architectures omitted it. In CNN 3 and 4, the first convolutional kernel was smoothed using a 2D Laplace filter ($3 \times 3$ pixels, weight 48) in the spatial domain and a 1D Laplace filter ($1 \times 3$ pixels, weight 0.02) in the temporal domain. Group B had no such smoothing or regularization. Each cell's RF location was modeled in the readout as an isotropic Gaussian with a learned mean. The predicted response was computed as the dot product of a learned weight vector and the shared core features at the learned location, followed by a parameterized softplus nonlinearity. For models within the first group, the vectors weighting the features for the modeled neurons were regularized using L1-norm with a coefficient of 0.01. Group B used no regularization.

**MEI optimization** We optimized spatio-temporal MEIs based on [28] using trained CNN model ensembles for each architecture type – 3, 4, 3B, and 4B. Each ensemble consisted of four trained CNNs of the same architecture but with different initializations. During MEI optimization, the weights of all models in the ensemble were frozen. For each modeled RGC $c$ and ensemble $a$ we synthesized an input $x_c$ such that $x_c = \arg\max_x f_a(x_c)$ where $f_a(x_c)$ is the ensemble's mean predicted activation for input $x$ for cell $c$. Optimization was performed using stochastic gradient descent from a Gaussian noise initialization (mean 0, standard deviation 0.01) with a learning rate of 5.0. We optimized the input for up to 40,000 epochs, stopping if the activation had not increased for 500 epochs. The inputs were constrained under an L2-norm of 3.5 to ensure that pixel values remained within the natural stimulus range of [-1, 1], avoiding clipping.

**Suppressive surrounds optimization** The suppressive surround optimization procedure was designed to synthesize a wide surround that drove wide-field inhibition. To achieve this, we followed a similar procedure to the MEI optimization with some modifications. For each modeled cell $c$ and ensemble $a$ we synthesized a surround $s_c$ such that $s_c = \arg\min_s f_a(x_c + (1-m)s)$ where $x_c$ is the MEI for cell $c$, $m$ is a mask protecting the previously obtained MEI from changing and $s$ is the input. The mask $m$ contained ones in diameter of 360 μm, centered on the center of the MEI, and zeros elsewhere. The diameter of 360 μm covered the RF size, including the standard surround, for all considered cell types. This setup allowed only pixels outside of the standard RF area to change and affect the ensemble's output during the optimization procedure.

## 5 Acknowledgements

This work was supported by the Deutsche Forschungsgemeinschaft (DFG, German Research Foundation) – project IDs 432680300 (SFB 1456, project B05) and 515774656 – and by the European Research Council (ERC) under the European Union's Horizon 2020 research and innovation programme (grant agreement number 101041669). Computing time was made available on the high-performance computers HLRN-IV at GWDG at the NHR Center NHR@Göttingen project nim00010.

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
