# A  Technical appendices and supplementary material

## A.1  CNN 3B and 4B performance and MEI generalization

We evaluated the performance of the group B architectures on the test set and found that they perform almost as well as the architectures from the initial group on both retinas (Figure 1A).

We also tested whether the MEIs generalize across the architectural groups. The results suggest they do. Group B architectures had a generally lower firing rate; however, the activation they elicited in response to the MEI generated from architectures from the initial group was comparable to their own. The same was also true the other way around. The MEIs generated on B-group architectures activated the initial CNNs similarly to their own MEIs (Figure 1B).

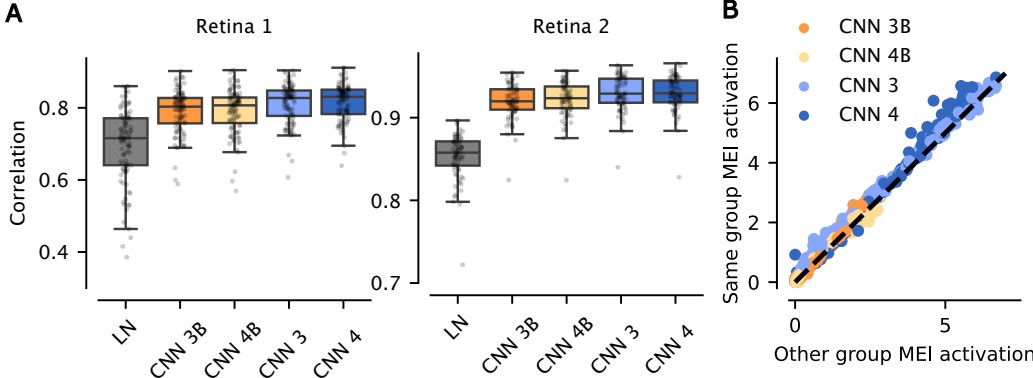

Figure 1: **A.** Prediction performance across models for Retina 1 (left) and Retina 2 (right). Boxplots show the distribution of correlation coefficients (CC) between model-predicted and recorded RGC responses. **B.** Comparison of MEI-elicited activation using MEIs generated from the same group as the model (y-axis) and the other group (x-axis) for all architectures. Specifically, each dot's y-value represents the average predicted firing rate to two MEIs – one generated by the architecture itself and one generated by the architecture within the same group. The x-value represents the average firing rate of the two MEIs generated by the architectures of the other group. Color indicates the architecture used to predict the firing rates.

## A.2  Additional cell type characteristics

We studied further characteristics of the cells – the cell-type specific distributions of explainable variance computed as in [1] (Figure 2A), the relationship between the mean firing rate of the cells and the cross-suppression (Figure 2B) and the temporal profiles of the MEIs taken from CNN 3, colored by cell type (Figure 2C).

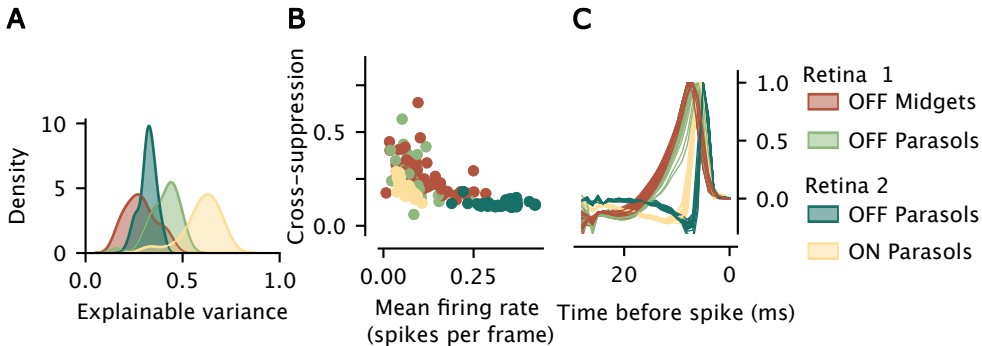

Figure 2: **A.** Reliability distributions of the different cell types across Retina 1 and Retina 2 **B.** Relationship between the mean firing rate of cells and their cross suppression. **C.** Temporal profiles of the MEIs generated using CNN 3 for all cells, colored by cell type.

### A.3 Single cell-type CNN performance

We trained all the CNN types on only cells from
each of our four cell types. The performance of these
CNNs compared to those trained on all cells did not
change (Figure 3).

### A.4 Cell selection and classification

We selected only cells that reliably responded to the
stimuli for the training of the models. The reliability
of the cells was determined based on the fraction of
explainable variance [1, 2] in each cell's responses
to the repeating segments of the stimuli across trials.
All cells that exceeded a threshold of 0.15 were used
for model training. This left us with 315 reliable cells
across *Retina 1* (235 cells) and *Retina 2* (80 cells).
To analyze the characteristics of RFs and suppressive
surrounds estimated by different models, we used the
cell-type classification provided with the retinal data
from [3]. In particular, we used the OFF midget (50
in Retina 1 and 0 in Retina 2), OFF parasol (31 in
Retina 1 and 38 in Retina 2), and ON parasol (0 in
Retina 1 and 35 in Retina 2) cell clusters.

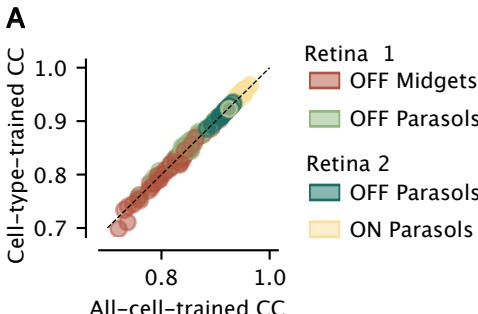

Figure 3: Comparison of correlation coeffi-
cients between models trained only using cells
of one specific cell type (y-axis) and models
trained using all reliable cells (x-axis). Each
dot represents one cell. The correlation value
for each cell was averaged over all four archi-
tectures: CNN 3, 4, 3B, and 4B.

### A.5 CNN training and evaluation

We trained the models using a Poisson loss between their predictions $p$ and responses $r$ on an 80/20
train/validation split of the non-repeating trials. We trained for up to 1,000 epochs with early stopping
triggered by no improvement in validation loss for 20 consecutive epochs. The initial learning rate
was 0.005, scheduled using PyTorch's ReduceLROnPlateau with a patience of 10 and a minimum
learning rate of 1e-8. Optimization was performed using the Adam algorithm [4].

We selected the final model architectures 3, 4, 3B, and 4B based on our requirements for distinct
hyperparameters, a wide spatial RF, and high validation correlation. The validation correlation was
estimated as an average correlation of four models of each architecture type on the validation part
of the training segment of the dataset. The reported performance is the final correlation coefficient
between the model predictions $p$ and the trial-averaged responses $\langle r \rangle$ to the held-out repeating test
sequence of the dataset.

Table 1: Hyperparameters of the four selected CNNs

| Architecture | Layers | Spatial kernels | Temporal kernels | Dilations | Channels | Grouped convolutions | Batch norm |
|---|---|---|---|---|---|---|---|
| CNN 3 | 3 | 29, 5, 5 | 25, 3, 3, 3 | 1, 1, 2 | 16, 32, 64 | 1 all | Yes |
| CNN 4 | 4 | 17, 5, 5, 5 | 27, 3, 3, 3 | 1, 1, 2, 3 | 8, 16, 32, 64 | 1 all | Yes |
| CNN 3B | 3 | 21, 7, 7 | 35, 3, 3, 3 | 1, 1, 2 | 16, 32, 64 | 16 all | No |
| CNN 4B | 4 | 21, 3, 3, 3 | 35, 3, 3, 3 | 1, 1, 1, 1 | 16, 32, 32, 64 | 16 all | No |

### A.6 LN models

To estimate the predicted activation of ensemble $a$ by an LN model's linear filter for cell $c$, we took
an LN model filter trained as described in [5]. Because the input into the CNN is a 3D array, but the
LN models had a separate spatial (2D) and temporal (1D) filter, we took the outer product of the
spatial and temporal filters to get a 3D version of the filter. Before multiplying, we smoothened the
spatial filter using a Kaiser window with $\beta = 5.0$. The LN model's softplus nonlinearity parameters,
including the threshold parameter $\delta$, were obtained from LN models from [3] as fitted to natural
movies.

## A.7 MEI size estimation

We used the generated MEIs to estimate the size of the modeled cells' RFs. We first separated the MEI using singular value decomposition into spatial and temporal components, ensuring the spatial filter was positive and the temporal filter was either positive or negative, depending on the polarity of the modeled cell. We used the spatial component to estimate the RF center and standard surround size. Specifically, we fit a Gaussian to the spatial filter with all negative values set to 0 and calculate the center size as $1.5\sigma$ where $\sigma$ is the diameter of a circle within a standard deviation of the fitted Gaussian. The surround size was estimated by thresholding pixel values averaged in radial distances around the mean of the fitted Gaussian. We considered radial distances that had averages below 0.3 times the minimal value of the filter as belonging to the surround, defining the surround size as the maximal radial distance that satisfied this condition. The reported RF sizes refer to the full MEI sizes, including the standard surround.

## A.8 Computational requirements

The CNN models were relatively small in terms of current deep learning standards. It took under 16 hours to train each of the CNN architectures on a single 40GB A100. It took under 3 minutes to generate a single MEI and under 5 minutes to generate a suppressive surround. It took around 1GB of memory to generate a single MEI/suppressive surround.