# OpenReview forum: "Interpreting convolutional neural networks to study wide-field amacrine cell inhibition in the retina"
_NeurIPS.cc/2025/Workshop/UniReps — UniReps2025_

### Official Review · Reviewer_Qnto · 2025-09-06
**The paper presents a novel application of CNN interpretability to study wide-field amacrine cell inhibition, revealing biologically plausible suppressive surrounds and generating hypotheses that could guide future experimental validation. While the motivation and biological framing are strong, the methodological novelty is limited, related work context is underdeveloped, and the organization could be improved for clarity.**

**Confidence:** 3

**Review:**

### Brief summary of paper
The paper uses convolutional neural networks (CNNs) trained on retinal ganglion cell (RGC) responses from marmoset retinas to probe the inhibitory role of wide-field amacrine cells (ACs). By combining most exciting input (MEI) optimization with a suppressive surround optimization procedure, the authors identify spatial patterns that mimic wide-field inhibition and reduce RGC activity by 10–30%. Results show both biologically plausible inhibitory surrounds and architecture-dependent limitations, suggesting CNN interpretability can generate testable hypotheses for retinal computation.

### PROS
- Clarity on impact statement/motivation question of the approach: "How interpretability methods applied to artificial neural networks can offer new hypotheses regarding biological retinal computation, which direct experimental recordings are hard, leading to a better targeted experimental validation."
- Originality on the "biological application" for utilizing Deep Learning model and its interpretability method (i.e., CNN model and its internal activations probing)

### CONS
- Lacking clarity (in-depth explanations) on:
  - 1. Related Work, focusing on more comprehensive Deep Learning approaches for understanding biological circuits
  - 2. Based on 1, what demarcates CNNs from other models. This would also allow readers to be more convinced why the baseline method was in the experiments was only LN.
- Limited novelty in terms of overarching methodology/approach
  - The core variation of CNNs seems to be the layer depth of the model (CNN3, CNN4).

- Organization of the paper: This is the minor stylistic aspect, which it would be more helpful to follow the results if the Methods (which was the last section of the paper) was layed out prior to the Results.

**Score:**

3

**Topic Fit:**

2

---

### Official Review · Reviewer_PfAv · 2025-09-15
**computational modeling using CNNs of the Amacrine cells**

**Confidence:** 3

**Review:**

This study used CNNs to model retinal responses and probe the role of wide-field amacrine cells (ACs). Different CNN architectures are trained for generalizability purposes.

This study provides valuable insights into wide-field amacrine cell function using convolutional neural networks (CNNs), but there are important considerations that the study is missing. While CNNs are a natural choice given that the retinal data were recorded from human adults, their architecture carries hardcoded inductive biases, akin to innate knowledge (such as weight sharing across layers, local receptive fields, spatial bias, etc.). This makes it hard to understand what is learned from experience and what is innate. Prior work has shown that even untrained CNNs can detect edges and recognize objects, raising questions about the extent to which their behavior truly reflects biological learning.

As a future direction, applying transformer-based models may offer a more flexible and powerful alternative, as they lack the spatial priors present in CNN-based models, making this line of research even more compelling.

**Score:**

4

**Topic Fit:**

3

---

### Official Review · Reviewer_wbWg · 2025-09-16
**Interpreting convolutional neural networks to study wide-field amacrine cell inhibition in the retina**

**Confidence:** 3

**Review:**

This study introduces a novel computational method to investigate the elusive inhibitory mechanisms of the retina by training a deep neural network as a functional proxy and then using a two-step optimization process to synthesize visual stimuli that reveal suppressive surrounds. While the method successfully identified inhibitory patterns that were robustly generalizable in one dataset, its failure to do so in a second highlights the critical importance of cross-architecture validation and caution when interpreting such models.

The work is of very high quality and originality, pioneering a rigorous, generalization-focused approach to computationally dissecting a latent neural mechanism that is presented with excellent clarity. However, the method's inconsistent performance is a significant weakness; the failure to produce generalizable surrounds for the Retina 1 dataset raises serious questions about the framework's reliability and the conditions required for its success.  Although this brittleness creates an unresolved ambiguity about whether non-generalizing features are model artifacts or biological realities, the paper's potential significance remains high, as it provides both a powerful new tool for generating testable hypotheses and a critical, cautionary tale for the field of computational neuroscience.

**On the Interpretation of Generalization Failure:** The failure of the suppressive surrounds to generalize for Retina 1 is a central, yet unresolved, finding.  Beyond the tested hypotheses of cell-type reliability and polarity diversity, do the authors have other speculations about the underlying cause? Could this failure point to a fundamental limitation of the core-readout CNN architecture in capturing the full dynamics of certain biological circuits, or could it be indicative of a non-stationarity or other problematic property within the Retina 1 dataset itself?

**On Biological Plausibility of Surround Structure:** The qualitative patterns of the suppressive surrounds are intriguing (e.g., the "loosely regular tiling" vs. "smoother surrounds"). Have you attempted to quantitatively characterize these structures or relate their spatial frequencies and temporal dynamics to known properties of specific AC subtypes? How do you interpret the consistent temporal offset, where the surround's peak intensity precedes the MEI's?

**Score:**

4

**Topic Fit:**

3

---

### Official Review · Reviewer_Zomq · 2025-09-16
**Novel approach to model the retina, solid background review and evidence**

**Confidence:** 3

**Review:**

Summary: The authors train convolutional neural networks to predict retinal ganglion cell (RGC) responses to natural movies, then use a two-step gradient-based feature visualization: first synthesize the Most Exciting Input (MEI) that maximally drives a target RGC, then freeze the MEI region and optimize the outside region to minimize the predicted response. The resulting outside pattern is interpreted as the model-inferred inhibitory surround; because the MEI/surround boundary is placed well beyond the RGC dendritic field, the authors argue this suppression reflects wide-field amacrine-cell influence. The approach is original and positioned as an in silico way to generate mechanistic hypotheses about long-range inhibition in retina.

Pros:
• Clear motivation and literature grounding: The manuscript is well-situated relative to prior retinal modeling and systems-neuroscience work.
• Careful experimental pipeline: The model fitting, model variants, and visualization pipeline are well engineered, predictive performance appears high and multiple architectures / datasets are evaluated.
• Novel, hypothesis-generating method: The two-step MEI→surround optimization is a clever way to probe long-range suppressive influences and has real potential to suggest experiments.

Cons:
• The authors should justify the design choice of repeating vs non-repeating stimuli during test and training phase respectively.
• Limited baseline comparisons: Performance comparison is currently to single-cell LN models only. Important alternative models with good performance, such as hierarchical LNs are not evaluated.
• Statistical reporting needs tightening: The manuscript reports group differences and correlations but does not consistently present effect sizes or appropriate statistical tests for correlations.

Quality: Technically strong, with a well-thought design and analysis.

Clarity: The manuscript is clearly written, with a good literature overview, and figures are informative.

Originality: The two-step MEI → surround feature-visualization applied to a fitted retinal CNN is a novel and creative idea that meaningfully extends feature-visualization methods into a hypothesis-generating context.

Significance: The approach seems to have high potential to guide experiments and advance understanding of long-range retinal inhibition.

**Score:**

4

**Topic Fit:**

2